# VAS3947 Induces UPR-Mediated Apoptosis through Cysteine Thiol Alkylation in AML Cell Lines

**DOI:** 10.3390/ijms21155470

**Published:** 2020-07-31

**Authors:** Maya El Dor, Hassan Dakik, Marion Polomski, Eloi Haudebourg, Marie Brachet, Fabrice Gouilleux, Gildas Prié, Kazem Zibara, Frédéric Mazurier

**Affiliations:** 1EA 7501 GICC, University of Tours, CNRS ERL7001 LNOx, Bâtiment Dutrochet, 10 boulevard Tonnellé, BP3223, CEDEX 1, 37032 Tours, France; mayaeldor@outlook.com (M.E.D.); hassan.dakik@mail.mcgill.ca (H.D.); marie.brachet@univ-tours.fr (M.B.); fabrice.gouilleux@univ-tours.fr (F.G.); 2PRASE, Beirut, Lebanon; kzibara@ul.edu.lb; 3EA 7501 GICC, University of Tours, IMT, 31 Avenue Monge, 37200 Tours, France; marion.polomski@etu.univ-tours.fr (M.P.); eloi.haudebourg@univ-tours.fr (E.H.); gildas.prie@univ-tours.fr (G.P.); 4Biology Department, Faculty of Sciences-I, Lebanese University, Beirut, Lebanon

**Keywords:** VAS3947, NADPH oxidases, leukemia, cysteine thiol alkylation, oxidative stress, endoplasmic reticulum, unfolded protein response

## Abstract

Nicotinamide adenine dinucleotide phosphate (NADPH) oxidases (NOX) involvement has been established in the oncogenic cell signaling of acute myeloid leukemia (AML) cells and in the crosstalk with their niche. We have shown an expression of NOX subunits in AML cell lines while NOX activity is lacking in the absence of exogenous stimulation. Here, we used AML cell lines as models to investigate the specificity of VAS3947, a current NOX inhibitor. Results demonstrated that VAS3947 induces apoptosis in AML cells independently of its anti-NOX activity. High-performance liquid chromatography (HPLC) and mass spectrometry analyses revealed that VAS3947 thiol alkylates cysteine residues of glutathione (GSH), while also interacting with proteins. Remarkably, VAS3947 decreased detectable GSH in the MV-4-11 cell line, thereby suggesting possible oxidative stress induction. However, a decrease in both cytoplasmic and mitochondrial reactive oxygen species (ROS) levels was observed by flow cytometry without disturbance of mitochondrial mass and membrane potential. Thus, assuming the consequences of VAS3947 treatment on protein structure, we examined its impact on endoplasmic reticulum (ER) stress. An acute unfolded protein response (UPR) was triggered shortly after VAS3947 exposure, through the activation of inositol-requiring enzyme 1α (IRE1α) and PKR-like endoplasmic reticulum kinase (PERK) pathways. Overall, VAS3947 induces apoptosis independently of anti-NOX activity, via UPR activation, mainly due to aggregation and misfolding of proteins.

## 1. Introduction

Acute myeloid leukemia (AML) are heterogeneous hematological malignancies, resulting from a block in the hematopoietic differentiation program, and characterized by a clonal expansion of myeloid blasts in the bone marrow (BM) and the peripheral blood (PB) [1]. Cancer cells exhibit a much higher level of oxidative stress compared to their normal counterparts. This has also been demonstrated for leukemic cells and, therefore, targeting the redox state in AML cells has been investigated [2]. Oxidative stress could be caused by an oncogenic signal, a change in energy metabolism, and/or a deficiency in the antioxidant system leading to higher production and/or accumulation of reactive oxygen species (ROS) [3]. Increasing evidence confirmed that tumor-derived ROS promote cell proliferation and survival [4], as well as metastasis [5,6] and drug-resistance [7]. Like the mitochondrial respiratory chain, nicotinamide adenine dinucleotide phosphate (NADPH) oxidases (NOX) are considered as major contributors of ROS production in cells and, hence, as interesting targets for therapy [2,3,8].

The NOX family comprises seven enzymatic complexes (NOX1-5, dual oxygenase (DUOX1)-2), that produce ROS as the main function [9]. NOX were found to display an effect on AML progression and chemoresistance [10,11,12,13,14]. In fact, AML proliferation is accompanied by NOX2-dependent extracellular ROS production in more than 60% of AML patients [10]. Studies have shown that fms-like tyrosine kinase 3—internal tandem repeat (FLT3-ITD) signaling involved NOX4 activity [15], while others demonstrated that NOX2 is implicated in leukemic development [11,16,17,18]. In addition, NOX-dependent ROS production has been shown to trigger mitochondrial transfer from BM stromal cells to AML blasts, thus fueling AML proliferation [11]. Furthermore, inhibition of NOX2 abolished this transfer and resulted in AML apoptosis and an improved survival rate of leukemia-bearing mice. In this context, we have recently characterized the expression of NOX subunits in various AML cell lines and cohorts of primary AML. Remarkably, our data highlighted that, except for NOX2 subunits, the expression of most NOX enzymes lacks in AML cells (Submitted manuscript). However, although diphenyleneiodonium (DPI), a widely used pan-NOX inhibitor, triggered apoptosis by inhibiting the respiratory chain reaction, the knockdown of *CYBB*, coding for NOX2, did not affect the proliferation and differentiation capacity of AML cell lines. Therefore, this implies that the interpretation of results could be biased by unspecific effects of NOX inhibitors, but also that AML cell lines could be used as models to validate the specificity of NOX inhibitors.

VAS3947, another NOX inhibitor, is a more soluble derivative of the VAS2870. Unlike DPI, VAS3947 does not seem to interfere with the flavoprotein XOD, the flavo-hemoprotein eNOS, or any ROS detection assay [19,20]. Although VAS3947 apparently blocks NOX after the complex is assembled to the membrane [21], its exact mechanism of action remains to be clarified. In this study, we evaluated VAS3947 specificity and mode of action in AML cell lines. We demonstrated that VAS3947 is able to target thiol on cysteine residues of glutathione and proteins, thus triggering apoptosis in AML cells through acute endoplasmic reticulum (ER) stress and cytotoxic signaling from unfolded protein response (UPR).

## 2. Results

### 2.1. VAS3947 Triggers Cell Proliferation Arrest and Death Independently of Anti-NOX Activity

To determine the cytotoxic effects of VAS3947, eight AML cell lines, covering M0 to M5 stages of the French–American–British (FAB) classification, were treated with increasing concentrations (0.5 µM to 20 µM) of this inhibitor for 72 h. The IC50 values of the different cell lines ranged from 2.6 ± 0.6 µM for the most sensitive cell line MV-4-11 to 4.9 ± 1.3 µM for THP-1, the least sensitive (Figure 1A). An average dose of 4 µM VAS3947 was chosen for the following experiments. After only 1 day-exposure at this concentration, all cell lines showed reduced cell numbers compared to their corresponding controls, and the difference increased over time (Figure 1B). Apoptosis was quantified at day 3 of exposure to investigate whether reduced cell numbers resulted from a decreased proliferation or a cell death induction. The VAS3947-induced decrease in cell proliferation was mostly explained by apoptosis for KG-1a, KG-1, ML-2, and MV-4-11 cells (Figure 1C). In contrast, VAS3947 triggered no or low apoptosis in HL-60, NB-4, U-937, and THP-1 at a 4 µM dose, while proliferation was slowed down. This can be explained by the fact that MV-4-11 cells received a VAS3947 dose equivalent to their IC90, while the THP-1 dose was much less than its IC50; hence, sufficient to induce cell cycle arrest but low apoptosis. NOX activity was not detected in the eight AML cell lines but could be stimulated in three of them, NB-4, ML-2, and THP-1 cells (Submitted manuscript). We confirmed the absence of NOX activity in THP-1 and its induction by phorbol 12-myristate 13-acetate (PMA), which, however, is efficiently inhibited by VAS3947 (Figure 2). Thus, VAS3947 reduces cell growth and induces cell death in the absence of NOX activity, thereby suggesting possible off-target effects.

### 2.2. VAS3947 Alkylates Cys Thiols of Glutathione (GSH) and Bovine Serum Albumin (BSA)

Since VAS2870 has been previously shown to alkylate thiol cysteine residues of GSH in vitro [22], we hypothesized that the VAS3947 effect on cells could likely originate from a similar mechanism. First, we assessed VAS3947 reactivity on the cysteine-containing tripeptide GSH in vitro using mass spectrometry (MS) and high-performance liquid chromatography (HPLC) analyses. The analysis of an admixture of VAS3947 and GSH revealed the appearance of a new species of 517.2 amu (atomic mass unit) corresponding to the alkylation product of GSH (307 amu) by the VAS3947 benzyltriazolopyrimidine moiety (210.2 amu) (Figure 3A). Remarkably, the MS analysis confirmed the lone presence of the VAS3947-GSH alkylation species (517.2 amu) in extracts from cells treated with 4 µM VAS3947 (Figure 3B). HPLC analysis of the admixture also revealed the appearance of the expected small oxazole-2-thiol leaving group along with the alkylation product (Figure 3C). It is worth noting that increasing GSH concentrations of admixtures showed a progressive loss of VAS3947, along with an increase in both the VAS3947-GSH alkylation compound and the oxazole-2-thiol leaving group, using both LC-MS and HPLC (Figure 3A,C). Thus, data indicate that VAS3947 alkylated thiol cysteine of GSH with its benzyltriazolopyrimidine moiety, leading to two new molecules, i.e., VAS3947-GSH adduct along with oxazole-2-thiol leaving group (Figure 3D).

To investigate the capacity of VAS3947 to alkylate free thiols on proteins in general, similar analyses were performed with BSA instead of GSH. BSA was chosen because of its wide use in biochemical applications and for being a well-known, small, and stable protein. MS results revealed the formation of a new species at 66638 Da, corresponding to a VAS3947-BSA alkylation compound (Figure 4A). As with the VAS3947-GSH admixture, HPLC analysis of the VAS3947-BSA admixture showed the appearance of both the oxazole-2-thiol leaving group and the alkylation product. In addition, increasing BSA concentration decreased the quantity of VAS3947 progressively and increased the oxazole-2-thiol leaving group proportion simultaneously (Figure 4B). Similar to GSH, results showed that VAS3947 alkylated free thiol cysteine of BSA with its benzyltriazolopyrimidine moiety, leading to VAS3947-BSA adduct and oxazole-2-thiol leaving group (Figure 4C).

### 2.3. Sensitivity to VAS3947 Inversely Correlates with the Glutathione Level in AML Cells

To explain the variability in cell sensitivities to VAS3947, we hypothesized that GSH levels could be different from one cell line to another. Quantification of GSH revealed high variabilities between the eight AML cell lines (Figure 5A). Remarkably, results showed that THP-1 cells, the least sensitive to VAS3947, had the highest levels of GSH (2.41 ± 0.02 mM/10^6^ cells), whereas MV-4-11 cells, the most sensitive to VAS3947, had the lowest GSH levels (0.34 ± 0.01 mM/10^6^ cells). However, although there is a trend towards a negative correlation between GSH level and VAS3947-induced apoptosis, this cannot be explained by GSH level alone; some cell lines showed similar levels of GSH and yet had variable sensitivities to VAS3947 (Figure 5B).

The difference between cell lines may be alleviated by the presence of antioxidant enzymes, and particularly those involved in the GSH system. Results at the transcriptional level showed that all cells were similarly equipped with antioxidant enzymes, especially MV-4-11 and THP-1, indicating that the GSH level is probably not explained by a difference in the antioxidant enzymes in these cells (Figure 5C). Furthermore, the effect of VAS3947 on intracellular total GSH was evaluated. After only 15 min of VAS3947 treatment, the level of total GSH in MV-4-11 cells dramatically decreased, whereas the decrease in THP-1 was not significant (Figure 5D). However, the differential (∆) decrease between MV-4-11 or THP-1 cells treated with VAS3947 and the corresponding control at 1 h was similar (∆_MV-4-11_ = 0.34 ± 0.04 vs. ∆_THP-1_ = 0.44 ± 0.26; *p* = 0.55). Thus, THP-1 kept a relatively high level of GSH, compatible with less induced cell death.

To determine whether GSH aggregation could explain induced cell death, the effect of VAS3947 in MV-4-11 cells was compared to that of *N*-ethylmaleimide (NEM). Results showed about a 2-fold lower efficacy of NEM in reducing cell proliferation, compared to VAS3947, with IC50 of 5.86 ± 1.70 vs. 2.6 ± 0.3 µM, respectively (Figure 5E). Therefore, a dose of 8 µM of NEM, corresponding to its IC90, was chosen to examine its effect on GSH level and apoptosis. Similar to VAS3947 at 4 µM, NEM at 8 µM triggered a drop of detected GSH (Figure 5F) and dramatic apoptosis (Figure 5G). Altogether, our results demonstrated that VAS3947 could induce cell death through cysteine thiol alkylation leading to depletion of free-GSH in AML cells.

### 2.4. VAS3947 Decreases ROS Levels

Since GSH has a potent antioxidant activity, a strong burst of ROS could be expected after VAS3947 treatment and GSH depletion, promptly leading to apoptosis. To examine this hypothesis, we assessed both cytoplasmic and mitochondrial ROS levels using 5-(and-6)-chloromethyl-2′,7′-dichlorodihydrofluorescein diacetate (CM-H2DCFDA) and MitoSOX probes, respectively. Notably, after 1 h of VAS3947 exposure, both cytoplasmic and mitochondrial ROS levels decreased strongly (Figure 6A,B). In addition, this instant decrease in mitochondrial ROS level did not depend on mitochondrial membrane potential (ΔΨm) or mass change (Figure 6C,D). It is worth noting that a slight disruption of the (ΔΨm) was observed after 2 h, which could denote a step toward the induction of apoptosis. Unlike VAS3947, NEM at 8 µM did not affect cytoplasmic ROS after 1 h of incubation (Figure 6E), while it was found to deplete glutathione and to induce apoptosis (Figure 5F,G). Unexpectedly, these results showed that apoptosis is not triggered by an increase in the oxidative stress due to antioxidant system blockage, but rather by a drop in the ROS level and/or aggregation of proteins, possibly leading to protein misfolding and ER stress. Remarkably, the pretreatment of MV-4-11 cells with 100 µM hydrogen peroxide (H_2_O_2_) did not rescue the cells from the cytotoxic effects of VAS3947, thus probably ruling out the hypothesis of apoptosis resulting from ROS decrease (Appendix A).

### 2.5. VAS3947 Triggers Endoplasmic Reticulum (ER) Stress and Consequent Unfolding Protein Response (UPR)

As demonstrated above, VAS3947 interacts with cysteine residues of proteins, which might cause an accumulation of misfolded proteins and, hence, provoke ER stress and UPR-induced apoptosis. To assess the effect of VAS3947 on ER stress, the expression of key players of the UPR system was analyzed. UPR is mediated by the activation of three pro-survival pathways, including transcription factor 6 (ATF6α), inositol-requiring enzyme 1α (IRE1α), and PKR-like endoplasmic reticulum kinase (PERK). Interestingly, VAS3947 did not affect the expression levels of both ATG12 and Beclin, thus ruling out the possibility of VAS3947-induced autophagy (Appendix A). We focused on IRE1α and PERK, well-known participants in inducing apoptosis. Indeed, PERK has been shown to inhibit the translation initiation factor 2α (eukaryotic initiation factor 2 alpha (eIF2α) by phosphorylation, thereby, reducing protein synthesis and overload [23,24,25]. In accordance, our results revealed rapid phosphorylation of eIF2α in MV-4-11 cells within 15 min following VAS3947 exposure (Figure 7A). This phosphorylation was maintained without any change in the expression level of eIF2α. On the other hand, IRE1α is known to induce the phosphorylation of c-Jun N-terminal protein kinase 1 (JNK1) and P38MAPK (mitogen-activated protein kinase), which then triggers apoptosis through the regulation of various pro- and anti-apoptotic proteins. Similar to eIF2α, our results showed rapid phosphorylation of JNK1 and P38MAPK after 15 min of VAS3947 treatment, which gradually increased over time (Figure 7B). Consequently, induction of caspase 9 and PARP cleavage was progressively observed after 60 min, indicating further an apoptotic mechanism (Figure 7C). Altogether, these data suggest that VAS3947-mediated aggregation leads to protein misfolding, which in turn results in ER stress and UPR-induced apoptosis.

### 2.6. Supplementation with GSH or N-acetyl Cysteine (NAC) Rescued MV-4-11 Cell Lines from VAS3947 Cytotoxic Effect

Our results demonstrated that VAS3947 has off-target effects by thioalkylating cysteine residues, whether on GSH or proteins, but whether that induced apoptosis should be further proven. Therefore, we investigated the effect of the supplementation with GSH or N-acetyl cysteine (NAC) on VAS3947-induced apoptosis. Remarkably, the MV-4-11 cells preincubation with GSH (1 mM), rescued the cells from the apoptotic effect of 4 µM VAS3947 (Figure 8A). NAC is a synthetic derivative of cysteine, and, as a precursor of GSH, allows GSH production only in the cells. The treatment of MV-4-11 cells with NAC (1 mM) increased the GSH level by 1.2 fold after 2 h (Figure 8B). This prevented the decrease in GSH level in cells treated with 4 µM VAS3947 (Figure 8C). Furthermore, NAC supplementation prevented MV-4-11 cells from the apoptotic effect of VAS3947 (Figure 8D) through inhibition of caspases activation triggered (Figure 8E). However, these results could be biased since VAS3947 could interact with the thiol groups of either GSH or NAC in the media, thus preventing their entry into the cells.

## 3. Discussion

In this study, we aimed to understand the mode of action of VAS3947 in AML cell lines, independently of its effect against NOX activity. We demonstrated for the first time that VAS3947 alkylates thiol cysteine residues of GSH in cells, as well as cysteine-free containing proteins, using BSA as an example. This interaction of VAS3947 led to two new molecules, GSH-VAS or Protein-VAS, along with a small oxazole-2-thiol leaving group. This oxazole-2-thiol group has already been observed when VAS2870 was incubated with GSH [22]. We did not observe any effect of oxazole-2-thiol on cell growth suggesting that the thiol alkylation of cysteine residues of GSH or proteins reduced cell proliferation via GSH depletion and/or unfolding of proteins (Figure 9).

Glutathione is a major contributor to the antioxidant system. The pool of reduced GSH is important for the functioning of other redox-modulating enzymes, such as peroxidases, peroxiredoxins, and thiol reductases. GSH is the most predominant thiol-containing peptide in mammalian cells, harboring a central cysteine residue. AML cells have an abundant expression of proteins implicated in the GSH system [23] and are highly sensitive to drugs targeting GSH metabolism [23]. Interestingly, the use of an enzyme that degrades cysteine, a precursor for the synthesis of GSH, completely eradicates leukemic stem cells (LSCs) without any effect on normal hematopoietic stem/progenitor cells [24]. Indeed, the authors showed that the survival of ROS^low^ LSCs is dependent on the availability of cysteine. The combination of arsenic trioxide (ATO) and homoharringtonine (HHT), drugs that perturb the mitochondrial redox state, with CB-839, a drug that blocks glutamine metabolism and consequently GSH production, efficiently induced cell death of AML cell lines, primary AML patient-derived samples, and in vivo mice models of AML [25]. Additionally, other studies showed that glutathione peroxidases (GPX)-1 and GPX-3, two antioxidant enzymes requiring GSH to detoxify hydrogen peroxide, are highly involved in AML development [24,26]. Therefore, aggregation of intracellular GSH by VAS3947, besides its inhibition of NOX-derived ROS production, may have a major effect in targeting the GSH system in AML. Remarkably, while we observed a dramatic drop of GSH level in MV-4-11 cells that have a low basal level, no induction of oxidative stress was detected, neither in the cytoplasm nor in mitochondria. In contrast, an unexpected decrease in ROS levels was shown. VAS2870 has been reported to decrease Nox4-derived ROS, involved in ischemic stroke induction [27]. Moreover, VAS2870 was shown to block the ROS production in liver cancer cells and their proliferation, which was attributed to NOX inhibition [28]. However, to our knowledge, no study has described the decrease in ROS for this molecule independently of the inhibition of NOX activity. It is well known that ROS are key molecules in cell signaling. Therefore, apoptosis could be explained by a decrease in ROS content, which is lower than the threshold required for cancer cell growth [29]. A recent study demonstrated an induced-apoptosis along with a dose-dependent decrease in ROS and GSH levels in hepatocellular carcinoma cells [30]. These results could be explained by an imbalance in the ROS level, which then promotes mitochondrial dysfunction and triggers mitochondria-mediated apoptosis. This mechanism of action should be investigated further to understand the complexity of ROS in cancer cells thoroughly. In addition, VAS3947 also interacts at the cysteine residues of proteins, which could possibly arrest cellular mechanisms, such as those regulating ROS homeostasis.

Protein aggregation in cells is known to induce ER stress and a cellular response through UPR. Thus, we hypothesized that the aggregation of VAS3947 with proteins may induce the accumulation of misfolded proteins, which could trigger ER stress and, consequently, UPR. The latter is a double-edged sword that promotes cell survival or apoptosis, depending on the capacity of the cell to regain redox homeostasis. UPR has been associated with three main ER sensor protein complexes: activating transcription factor 6 (ATF6), PKR-like endoplasmic reticulum kinase (PERK), and inositol-requiring enzyme 1α (IRE1α) [31,32,33]. In homeostatic conditions, these three factors are sequestered in the ER lumen by the interactions with binding immunoglobulin protein (BiP), which retains them in an inactive state. Following the induction of ER stress, BiP dissociates and binds unfolded proteins, which causes the ER sensors to dimerize and induce their autophosphorylation and activation [34]. Next, BiP directs the misfolded proteins to degradation through the ER-associated degradation (ERAD) complex. Our results showed that exposure to VAS3947 leads to rapid phosphorylation of P38MAPK and JNK1, downstream of the IRE1α activation, as well as the phosphorylation of eIF2α, downstream of PERK activation. However, Lu et al. recently showed that VAS3947 inhibits the activation of P38MAPK, reducing platelet activation and thrombus formation [35], a difference that could be explained by the different cell types or signaling pathways. It is noteworthy that the authors also demonstrated that VAS3947 acts independently of NOX inhibition downstream of PKC activation. More remarkably, the effect obtained on thrombus formation was achieved without any impact on normal cell hemostasis in mice, which provides a proof of principle for the safe administration of VAS3947 in vivo. Given that accumulating evidence has shown that the UPR is involved in the pathogenesis of AML [36,37], recent studies aimed to inhibit this mechanism to target AML cells [38,39]. For instance, these studies demonstrated that UPR activation triggered apoptosis in AML cells. In fact, the treatment of AML cells with IRE1α inhibitors increased the levels of miR-34a, thus causing a cell cycle arrest in the G1 phase and apoptosis [40]. In addition, the silencing of HIF-2α, which caused an increase in ER stress and ROS production, triggered apoptosis in AML cells [41]. Moreover, the treatment of patients’ AML cells with ONC201, an inducer of p53-independent apoptosis, promoted ATF4-mediated apoptosis [42]. Finally, a recent study also showed that the combination of drugs that induce ER oxidative stress and the differentiation agent retinoic acid (RA) could be an effective therapy to target AML cells characterized by FLT3-ITD mutations [43]. However, for the first time, we show that targeting the UPR is accompanied by ROS level drop, whereas to date, it is well known that triggering the ER stress is followed by a drastic increase in ROS [38,44], thus proposing new mechanisms which should be elucidated. Therefore, since VAS3947 triggers cell apoptosis through activation of the UPR downstream pathway, further investigation is necessary to determine its efficacy in combination with current AML treatment with ARA-C, particularly on cells that are resistant to conventional therapy.

## 4. Materials and Methods

### 4.1. Solution Preparation

VAS3947 (Merck Millipore, Fontenay-sous-Bois, France) stock solution was made at 10 mM in DMSO. Glutathione (GSH), bovine serum albumin (BSA) and *N*-ethylmaleimide (NEM) stock solutions were made at 10 mM in water. For mass spectrometry (MS) and high-performance liquid chromatography (HPLC), GSH (10, 20 and 40 µM) and BSA (10, 20 and 40 µM) were separately incubated with a two-fold excess of VAS3947 (20 µM) in PBS for 20 min at room temperature before analysis. Oxazole2-thiol (14 µM) was used as control. All solutions were purchased from Sigma–Aldrich (Saint-Quentin-Fallavier, France).

### 4.2. Cell Lines and Culture

Eight human acute myeloid leukemia cell lines (KG-1a, KG-1, HL-60, NB-4, ML-2, THP-1, MV-4-11, U-937) covering M0 to M5 FAB stages were purchased from DSMZ (German Collection of Microorganisms & Cell Cultures, Braunschweig, Germany), used at a maximum of 20 passages, and frequently tested for the absence of mycoplasma contamination. Cells were cultured in RPMI media, supplemented with 10% fetal bovine serum (FBS), at 37 °C in fully humidified air and 5% CO_2_. Cells were harvested from the culture at the exponential growth phase.

### 4.3. Cell Number Measurement

Cells were seeded at a density of 4 × 10^3^ cells/well in 100 μL of RPMI media and incubated overnight at 37 °C. Cells were then exposed to various concentrations of VAS3947 in a final volume of 200 μL. Cultures were performed 3 days after the addition of VAS3947. Cell proliferation was followed at the indicated time points by the Resazurin fluorescence assay (Sigma–Aldrich). Briefly, Resazurin (0.1 mg/mL) was added at 20 μL/well and incubated for 4 h at 37 °C in the dark, then fluorescence (λ_ex_ = 529.5 ± 19 nm, λ_em_ = 582 ± 36 nm) was measured using the ClarioStar microplate reader (BMG Labtech, Champigny-sur-Marne, France).

### 4.4. Apoptosis Assay

Cells were cultured alone or in the presence of VAS3947 (4 μM), as described above. Three days following drug addition, cells were harvested and washed with cold PBS and then resuspended in Annexin V binding buffer (BioLegend, London, UK). Next, cells were stained with APC-conjugated Annexin V (BioLegend) and 7-AAD (Sigma–Aldrich) according to the manufacturer instructions and analyzed using a C6 Accuri^®^ flow cytometer (Becton Dickinson, Le Pont de Claix, France) and FlowJo^®^ software (Becton Dickinson).

For GSH experiments, cells were first incubated with GSH (1 mM) for two hours, then washed twice with warm media and treated with 4 µM VAS3947 for different time points (2, 6, 24, 48, 72 h). Cells were then harvested and washed with PBS, followed by resuspension in Annexin V binding buffer. Next, cells were stained with APC-conjugated Annexin V according to the manufacturer’s instructions and analyzed using a C6 Accuri^®^ flow cytometer (Becton Dickinson,) and FlowJo^®^ software (Becton Dickinson).

### 4.5. Enzymatic Activity Assay

Cells were washed in PBS and seeded in 96-well plates at 2 × 10^5^ cells per well in isotonic glucose buffer (300 mM) containing luminol (10 µM) (Sigma–Aldrich) and horseradish peroxidase (10 U/well) (Sigma–Aldrich). Phorbol myristate acetate (PMA) (Sigma–Aldrich) was injected at 100 nM final concentration before measurement. Emitted luminescence was measured every minute over 2 h at 25 °C on a ClarioStar microplate reader (BMG Labtech). The area under the curve was calculated via ClarioStar Data Analysis software^®^ from average over replicates of blank-corrected data. VAS3947 was used at 4 µM as a pan-NOX inhibitor, and DMSO (Sigma–Aldrich) was used as vehicle control.

### 4.6. Mass Spectrometry (MS)

HRMS experiment was realized using the Acquity UPLC H-Class system hyphenated to a Vion IMS QTof mass spectrometer (both from Waters, Wilmslow, UK).

#### 4.6.1. For In Vitro VAS3947, GSH and BSA

Before MS analysis, 1 ng of the sample was injected onto a BEH C18 2.1 × 50 mm, 1.7 µm column heated to 50 °C. A 6 min gradient from 5% to 90% solvent B was applied with 0.5 mL/min flow rate to elute the sample (solvent A: H_2_O + 0.1% Formic Acid, solvent B: Acetonitrile + 0.1% Formic Acid). MS data were acquired using positive ionization mode with an ESI source over a 50 to 1400 *m*/*z* window with a 0.2 Hz scan rate and collision energy ramp from 20 eV to 40 eV. Voltage capillary was set to 1.5 kV, desolvation temperature and source temperature to 600 and 120 °C, respectively. Data were processed using UNIFI software version 1.9.4 (Waters, Wilmslow, UK).

#### 4.6.2. For Cell Extracts

Before MS analysis, 800 ng of the sample was injected onto a BEH C4 2.1 × 30 mm, 1.7 µm column heated to 90 °C. A desalting step was then carried, with 95% solvent A (H_2_O + 0.1% Formic Acid) and 5% solvent B (Acetonitrile + 0.1% Formic Acid) during 2 min at 0.5 mL/min, with the flow diverted to waste. Then, a 4 min gradient from 5% to 90% solvent B was applied with 0.4 mL/min flow rate to elute the sample with the flow diverted to MS. MS data were acquired using positive ionization mode with an ESI source over a 500 to 4000 *m*/*z* window with 1 Hz scan. Voltage capillary was set to 2.5 kV, desolvation temperature and source temperature to 600 and 120 °C, respectively, and cone voltage at 150 V. Results were processed using UNIFI software version 1.9.4 and MaxEnt1 algorithm for deconvolution.

### 4.7. High-Performance Liquid Chromatography (HPLC)

HPLC analyses were carried out with a LaChrom Elite system (Hitachi L-2130 (pump) and L-2400 (UV-detector)) using 254 nm UV for detection. The column was a Phenomenex Synergi™ C-12, 4 µm particle size (100 mm × 4.6 mm), supported by a Phenomenex Security Guard cartridge kit C12 (4.0 mm × 3.0 mm). Elution was performed with 0.2% (by volume) of TFA in water (solvent A), and acetonitrile (solvent B); gradient 20–100% of B with a flow rate of 1 mL/min; column temperature of 25 °C; injection of 50 µL in DMSO or PBS (pH 7.2).

### 4.8. Glutathione Measurement

Total glutathione (GSH), reduced and oxidized, was determined by the GSH cyclic reductase assay, as described by Tietze [45]. A minimum of 10^6^ cells was washed with PBS, followed by the resuspension in 5% 5′-sulfosalicylic acid (SSA) solution and seeded in a 96-well plate for GSH analysis. A mixture of ethylenediamine tetraacetic acid (EDTA, 0.95 mM), potassium phosphate buffer (95 mM), glutathione reductase (0.15 units/mL) and Ellman’s reagent 5,5′-dithio-bis-2-nitrobenzoic acid (DTNB, 78 µM) was added per well resulting in a 180 μL reaction volume. Nicotinamide adenine dinucleotide phosphate (NADPH, 48 µM) was instantly injected per well, and the absorbance was measured during 5 min at a wavelength of λ = 412 nm using the ClarioStar microplate reader. *N*-ethylmaleimide (NEM) was used as a positive control for GSH aggregation. Total GSH is expressed in mM per 10^6^ cells.

### 4.9. Real-Time Reverse Transcription Quantitative PCR (RT-qPCR) Assay

RT-qPCR reactions were performed in a total volume of 10 μL on 20 ng of cDNA using LightCycler^®^ 480 Probes Master (Roche, Meylan, France). Samples were subjected to initial denaturation step (5 min, 95 °C), followed by 45 PCR cycles (10 s, 95 °C, then 30 s, 60 °C) and a final cooling step (40 °C, 30 s). All reactions were run in triplicates, and results analyzed using the Cycle threshold (Ct) values determined with the LightCycler^®^ 480 software. The primer sequences used for qRT-PCR are cited in Picou et al. [46]. The geometric Ct mean of human ACTB, YWHAZ, and RPL13A were used as an endogenous control to normalize the expression of target genes: ΔCT = “Ct target”−“Ct reference geomean”.

### 4.10. ROS Measurement

Cells were washed with PBS and were resuspended at 10^5^ cells in 50 μL reaction volume. 5-(and-6)-chloromethyl-2′,7′-dichlorodihydrofluorescein diacetate (CM-H2DCFDA) and MitoSOX (both from Invitrogen, Villebon-sur-Yvette, France) were respectively used at a final concentration of 5 μM to measure cytoplasmic ROS level and mitochondrial superoxide production. Cells were incubated in the dark at 37 °C for 30 min. After treatment with VAS3947 at different time points, the fluorescence of CM-H2DCFDA or MitoSOX was measured at λ_em_ = 530 ± 30 nm and λ_ex_ = 489 ± 14 nm, respectively, on a C6 Accuri**^®^** flow cytometer. Data were analyzed using FlowJo**^®^** software.

### 4.11. Mitochondrial Membrane Potential Measurement

Cells were washed with PBS and resuspended at 10^5^ cells in 50 μL reaction volume. To measure mitochondrial membrane potential (ΔΨ_m_), tetramethyl-rhodamine ethyl ester (TMRE; Invitrogen) was added at a final concentration of 5 μM. Cells were incubated in the dark at 37 °C for 30 min before fluorescence measurement. After treatment with VAS3947 at different time points, the fluorescence of TMRE was measured at λ_em_ = 580 ± 25 nm and λ_ex_ = 550 ± 15 nm on a C6 Accuri**^®^** flow cytometer. Data were analyzed using FlowJo**^®^** software.

### 4.12. Mitochondrial Biomass Measurement

To measure mitochondrial biomass, cells were washed with PBS and resuspended at 10^5^ cells in 50 μL reaction volume. MitoTracker red was added (Invitrogen) at a final concentration of 50 nM. Cells were incubated in the dark at 37 °C for 30 min. After treatment with VAS3947 at different time points, the mitochondrial biomass was measured by the fluorescence at λ_em_ = 594 ± 25 nm and λ_ex_ = 580 ± 15 nm on a C6 Accuri**^®^** flow cytometer. Data were analyzed using FlowJo**^®^** software.

### 4.13. Protein Extraction and Western Blotting Assay

Proteins were extracted using Laemmli blue buffer (62.5 mM Tris-HCl pH 6.8, 2% SDS, 10% glycerol, 5% β-mercaptoethanol, 0.005% bromophenol blue). Extracted proteins were heated at 95 °C for 10 min before loading. Protein extracts from 5 × 10^5^ cells were loaded per well in a 4–15% gradient polyacrylamide gel (BioRad, Marnes-la-Coquette, France) and separated by SDS-PAGE for 25 min at 200 V. Proteins were then transferred onto a 0.2 μm nitrocellulose membranes (BioRad) and blocked for 1 h at room temperature in 0.2% TBS-Tween buffer containing 5% milk. Afterward, membranes were incubated overnight at 4 °C with appropriate dilutions of primary antibodies prepared in TBS-Tween solution with 5% milk. The primary antibodies used were purchased from Cell Signaling Technology (CST) and diluted as recommended at 1/1000: rabbit anti-P38MAPK (#9212CST), rabbit anti-p-P38MAPK (#4511), mouse anti-JNK1 (#3708), rabbit anti-p-JNK1 (#4668), rabbit anti-eIF2α (#5324), rabbit anti-p-eIF2α (#3398), rabbit anti-caspase 9 (#9502), rabbit anti-PARP (#9532), rabbit anti-ATG12 (#4180CST), rabbit anti-Beclin (#3495). The mouse anti-β-actin (SC-47778) was purchased from SantaCruz. Primary antibodies were diluted as recommended at 1/1000. Secondary HRP-conjugated anti-mouse and anti-rabbit antibodies (Vectorlabs, Peterborough, UK), diluted at 1/10,000, were used for band detection via the Amersham ECL detection kit (GE Healthcare Life Sciences, Velizy-Villacoublay, France). Secondary HRP-conjugated anti-mouse and anti-rabbit antibodies (Vectorlabs, Peterborough, UK) were used for band detection via the Amersham ECL detection kit (GE Healthcare Life Sciences, Velizy-Villacoublay, France).

### 4.14. Statistical Analysis

Results are expressed as mean ± SEM of at least 3 independent experiments. GraphPad Prism software^®^ (Version 6.01, San Diego, CA, USA) was used to perform statistical analysis. Significance is indicated by *, *p* < 0.05; **, *p* < 0.01; and ***, *p* < 0.001.

## 5. Conclusions

In summary, this study shows for the first time that VAS3947 induces apoptosis in AML cell lines independently of its anti-NOX activity, while it also draws attention to a possible bias in data interpretation concerning NOX implication in AML cells. However, VAS3947 would be worth testing as a multitarget molecule based on its properties of inhibiting NOX activity, blocking the antioxidant system, and inducing protein ER stress; three cellular mechanisms that have been previously explored to target AML cells.

## Figures and Tables

**Figure 1 ijms-21-05470-f001:**
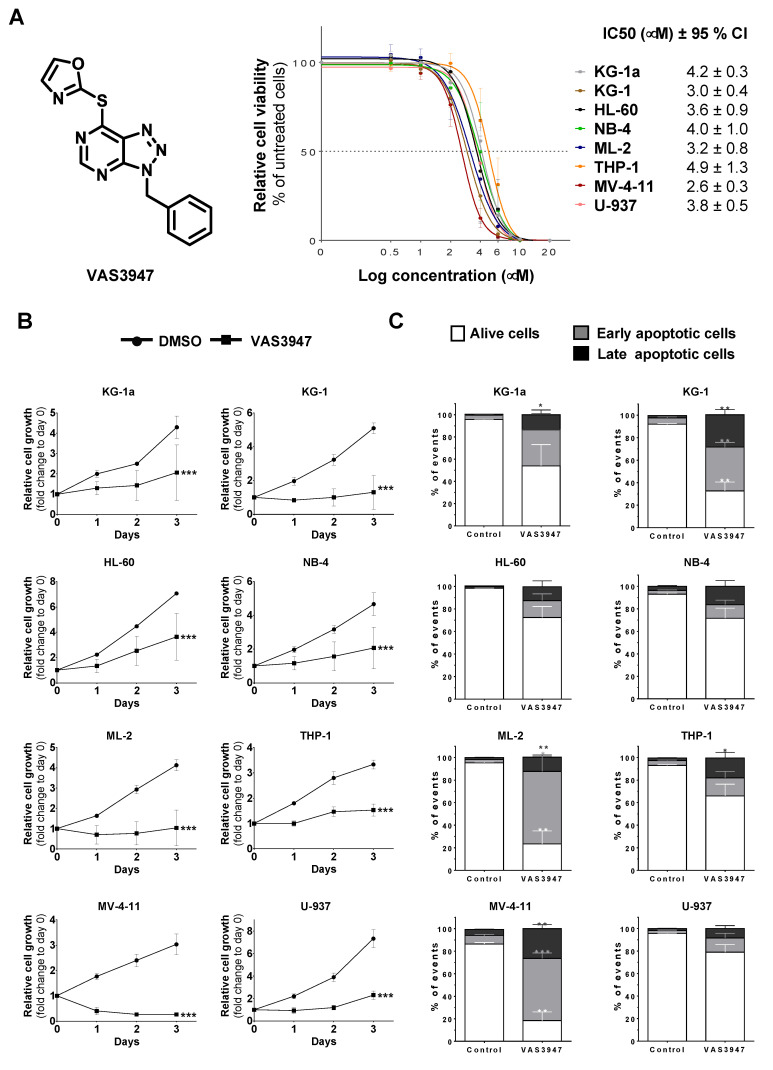
Effect of 4 µM VAS3947 on cell growth. (**A**) Dose-response curves of VAS3947 for eight acute myeloid leukemia (AML) cell lines, as indicated, and the corresponding IC50 (in µM) ± 95% confidence interval (CI). (**B**) Effect of VAS3947 on the growth of eight AML cell lines. Cell proliferation was assessed using a resazurin reduction assay at indicated days following VAS3947 treatment. Relative cell growth was calculated as resazurin fluorescence fold change compared to control at day 0. Data are shown as mean ± SEM of *n* = 3 independent experiments. Two-way ANOVA was performed for each cell line, followed by Tukey’s post hoc analysis. Adjusted *p*-values are obtained by comparing the VAS3947-treated condition to the DMSO-treated control. (**C**) Apoptosis in VAS3947-treated AML cell lines. Analysis of apoptosis was performed on day 3 following treatment. Data are shown as mean ± SEM (*n* = 3). Student’s *t*-test was performed for each cell line, comparing various proportions from VAS3947 conditions to their corresponding control counterparts (* *p* < 0.05; ** *p* < 0.01; *** *p* < 0.001).

**Figure 2 ijms-21-05470-f002:**
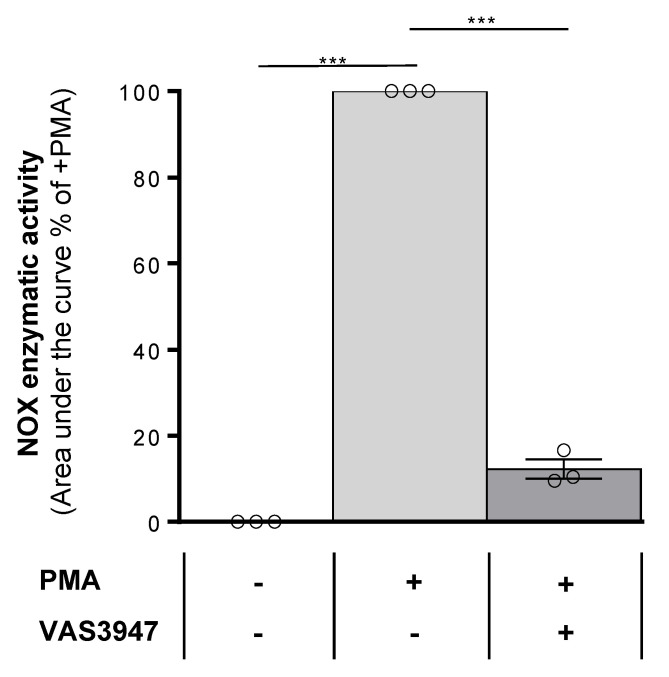
Effect of VAS3947 on phorbol 12-myristate 13-acetate (PMA)-induced NOX activity in THP-1 cells. PMA injection induced NOX activity in THP-1 cells, while the addition of VAS3947 at 4 µM blocked most of this induction. Results are shown as the mean ± SEM of *n* = 3 independent experiments. Student’s *t*-test was performed comparing each treated condition to the corresponding control counterpart (*** *p* < 0.001).

**Figure 3 ijms-21-05470-f003:**
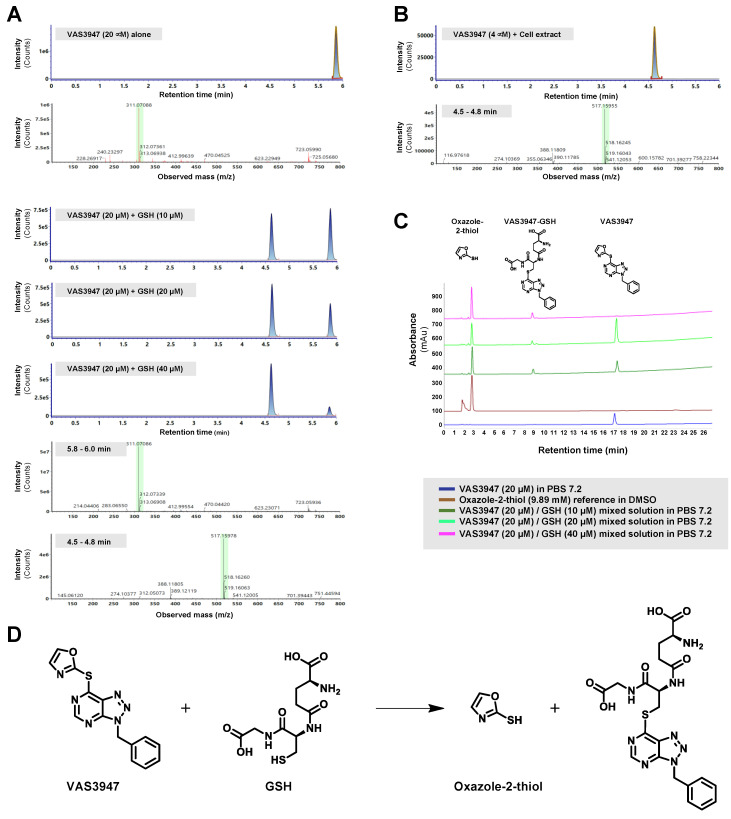
LC-ESI MS and high-performance liquid chromatography (HPLC) analysis of the glutathione (GSH) modification by VAS3947. (**A**,**B**) Extracted ion current chromatograms and corresponding mass spectra are shown for VAS3947, VAS3947/GSH mixtures at indicated concentrations (20 min incubation), and the cell extracts pretreated with 4 µM VAS3947. (**C**) HPLC analysis for VAS3947, the oxazole-2-thiol, and VAS3947/GSH mixtures at various concentrations. (**D**) MS and HPLC analyses indicate that the thiol function of GSH is alkylated by the benzyltriazolopyrimidine moiety of VAS3947, with the oxazole-2-thiol moiety of VAS3947 serving as a leaving group.

**Figure 4 ijms-21-05470-f004:**
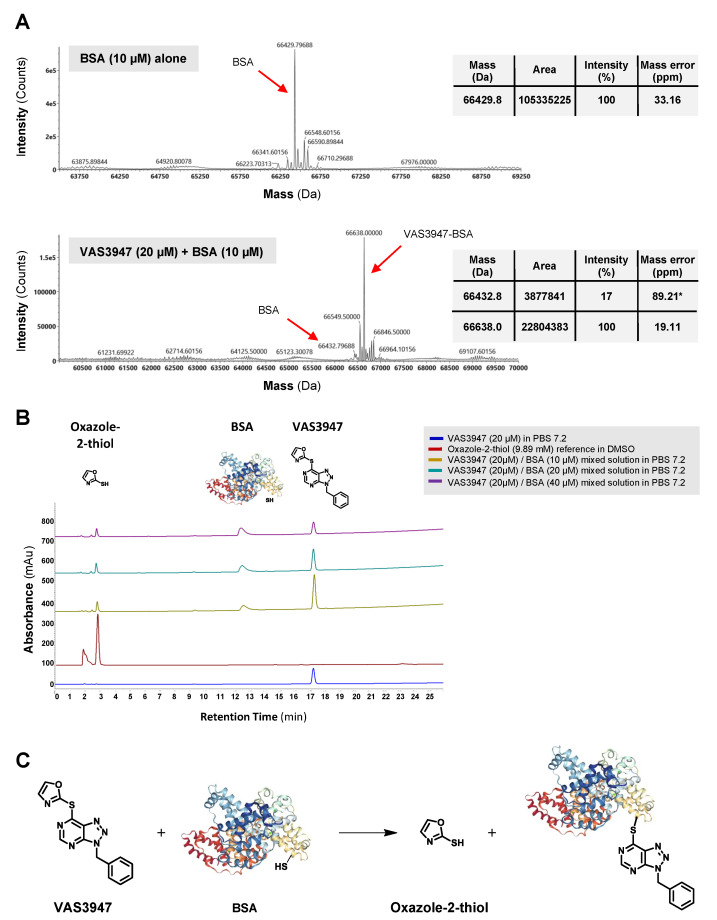
LC-ESI MS and HPLC analysis of the BSA modification by VAS3947. (**A**) Extracted ion current chromatograms and corresponding mass spectra are shown for VAS3947 and a VAS3947/BSA mixture (20 min incubation). * Low intensity observed for BSA makes it difficult for the MaxEnt1 algorithm to catch a mass with high precision for this protein. (**B**) HPLC analysis for the same solution of VAS3947, oxazole-2-thiol, and the VAS3947/BSA mixture. (**C**) MS and HPLC analyses indicate that the thiol function of BSA is alkylated by the benzyltriazolopyrimidine moiety of VAS3947, with the oxazole-2-thiol moiety of VAS3947 serving as leaving group.

**Figure 5 ijms-21-05470-f005:**
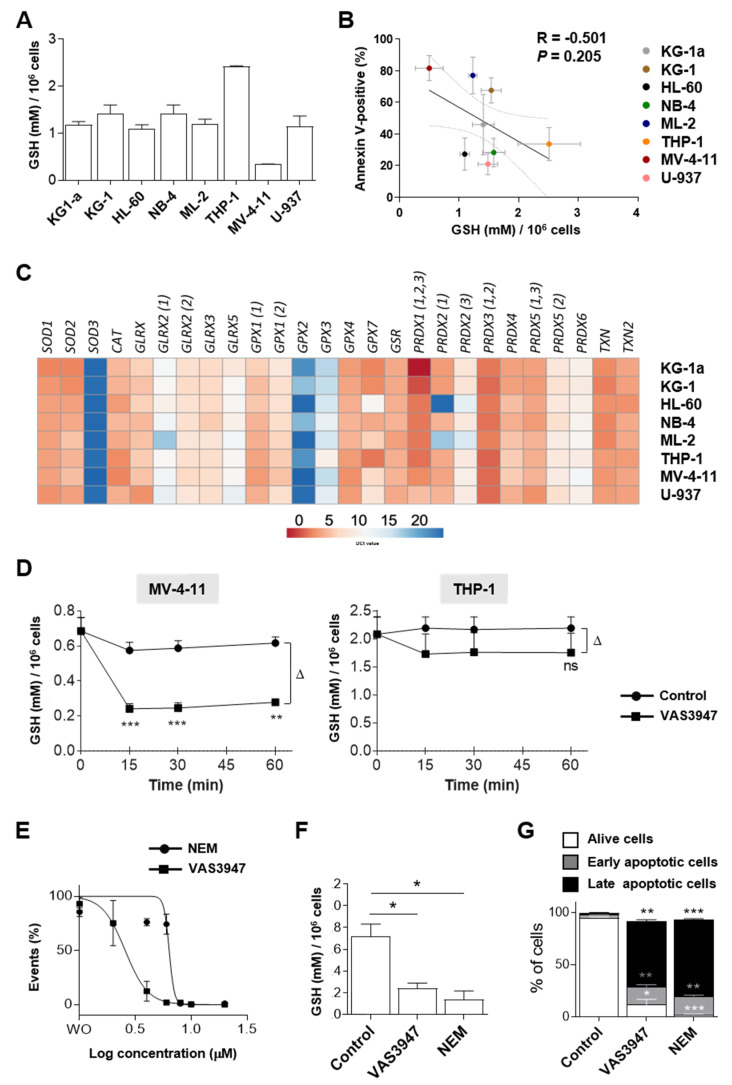
Evaluation of physiological glutathione (GSH) and the effect of 4 µM VAS3947 on AML cell lines. (**A**) Levels of GSH in eight AML cell lines. GSH levels were detected based on Tietze’s method. Data are shown as mean ± SEM of *n* = 3 independent experiments. (**B**) Correlation between GSH levels corresponding to data presented in A and apoptosis level corresponding to data presented in Figure 1C in AML cell lines. (**C**) mRNA expression heatmap of 22 antioxidants assessed by RT-qPCR in the eight AML cell lines. Results are shown as the mean ∆Ct ± SEM of *n* = 3 independent experiments. Colors reflect expression levels as ∆Ct mean values (red: high expression, blue: low expression). Numbers within parentheses indicate detected gene variants. (**D**) Effect of 4 µM VAS3947 on intracellular GSH in MV-4-11 and THP-1 cell lines. Data are shown as mean ± SEM of *n* = 3 independent experiments. (**E**) Dose-response curve of VAS3947 and *N*-ethylmaleimide (NEM) for the MV-4-11 cell line and their IC50 (in µM) ± 95% CI. Data are shown as mean ± SEM of *n* = 3 independent experiments. (**F**) Comparing the GSH level in MV-4-11 cell line treated with either VAS3497 (4 µM) or NEM (8 µM). Data are shown as mean ± SEM of *n* = 3 independent experiments. (**G**) Comparing the apoptotic effects of VAS3497 (4 µM) and NEM (8 µM) on the MV-4-11 cell line. Data are shown as mean ± SEM of *n* = 3 independent experiments). Student’s *t*-test was performed for each cell line, comparing each treated condition to its corresponding control counterpart of individual time points (* *p* < 0.05; ** *p* < 0.01; *** *p* < 0.001). Δ: difference.

**Figure 6 ijms-21-05470-f006:**
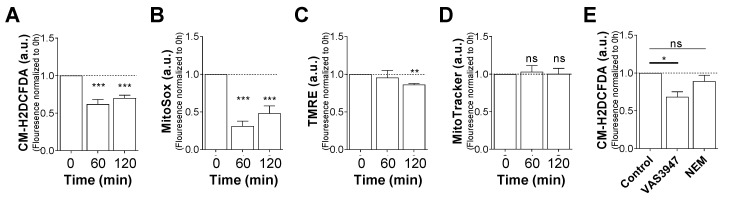
Effect of VAS3947 on cytoplasmic and mitochondrial reactive oxygen species (ROS) production in MV-4-11 cell line. Effect of VAS3947 (4 μM) on (**A**) cytoplasmic ROS production measured by 5-(and-6)-chloromethyl-2′,7′-dichlorodihydrofluorescein diacetate (CM-H2DCFDA) fluorescence; (**B**) mitochondrial ROS production measured by MitoSOX fluorescence; (**C**) mitochondrial membrane potential measured by tetramethyl-rhodamine ethyl ester (TMRE) fluorescence; and (**D**) mitochondrial biomass measured by MitoTracker fluorescence. (**E**) Effect of VAS3947 (4 μM) and NEM (8 μM) on cytoplasmic ROS production measured by CM-H2DCFDA fluorescence. ROS measurement at 0 h (**A**–**D**) and control (**E**) was determined and established at 1. Data are shown as normalized fluorescence with respect to the control. Data are shown as mean ± SEM of *n* = 3 independent experiments. Student’s *t*-test was performed comparing VAS3947 conditions to control (* *p* < 0.05; ** *p* < 0.01; *** *p* < 0.001).

**Figure 7 ijms-21-05470-f007:**
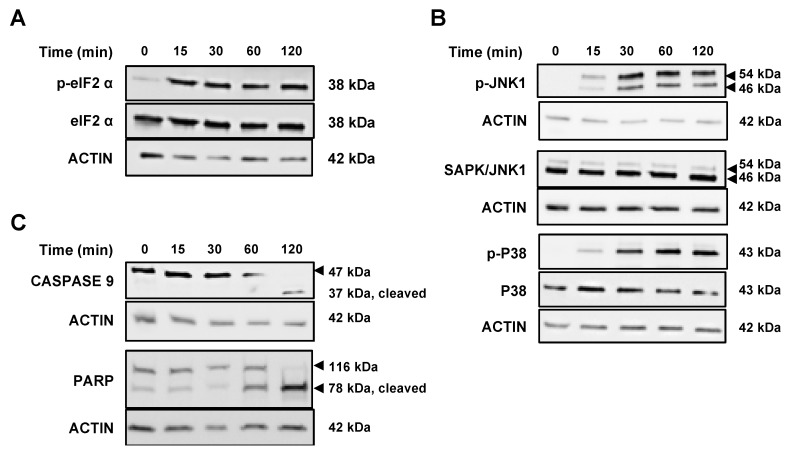
Effect of VAS3947 on unfolded protein response (UPR) proteins and apoptosis in MV-4-11 cell lines. (**A**,**B**) Western blot analysis showing the effect of VAS3947 (4 µM) on the expression of UPR proteins and their phosphorylation status at different indicated time points. (**A**) The activation of the PKR-like endoplasmic reticulum kinase (PERK) pathway is shown by the phosphorylation of eukaryotic initiation factor 2 alpha (eIF2α). (**B**) The activation of inositol-requiring enzyme 1α (IRE1α) is shown by the phosphorylation of c-Jun N-terminal protein kinase 1 (JNK1) and P38MAPK (mitogen-activated protein kinase). (**C**) Western blot analysis showing the activation of apoptosis through caspase 9 and PARP proteins’ cleavage at different indicated time points after VAS3947 (4 µM) treatment. The β-actin was used as loading control.

**Figure 8 ijms-21-05470-f008:**
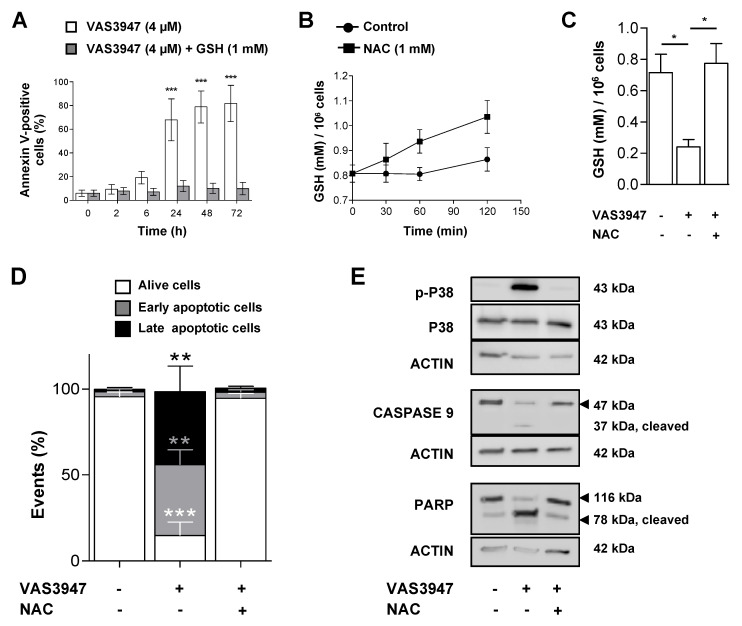
Glutathione (GSH) or N-acetyl cysteine (NAC) supplementation prevents of MV-4-11 cells from VAS3947-induced apoptosis. (**A**) MV-4-11 cells were preincubated with GSH (1 mM) for 2 h, washed, and incubated with VAS3947 prior annexin V-positive cell detection by flow cytometry at the indicated timepoints. Data are shown as mean ± SEM of *n* = 3 independent experiments. (*** *p* < 0.001) (**B**) GSH production measured over time in MV-4-11 cells incubated with NAC (1 mM). Data are shown as mean ± SEM of *n* = 3 independent experiments. (**C**) Effect of VAS3947 (4 µM) on intracellular GSH level in the presence or absence of a 2 hour-NAC preincubation in MV-4-11 cells. Data are shown as mean ± SEM of *n* = 3 independent experiments. Student’s *t*-test was performed to compare various conditions to their corresponding control counterparts (* *p* < 0.05). (**D**) Effect of NAC (1 mM) supplementation on VAS3947 (4 µM)-induced apoptosis in MV-4-11 cells. Student’s *t*-test was performed for each condition, comparing them to their corresponding control counterparts (** *p* < 0.01; *** *p* < 0.001). (**E**) Western blot analysis of apoptotic protein expressions in MV-4-11 cells after treatment as depicted in Figure 8D.

**Figure 9 ijms-21-05470-f009:**
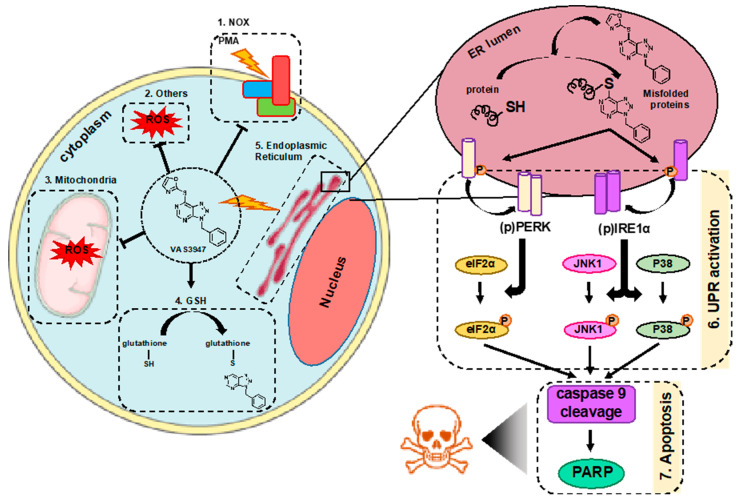
Schematic representation of the effect of VAS3947 on AML cell lines. In AML cell lines, beside its effect as an inhibitor of NOX-derived ROS production (1), VAS3947 has other off-target effects. VAS3947 induces NOX independent cytoplasmic ROS decrease (2) as well as mitochondrial ROS production (3) by a yet unclear mechanism. Moreover, it efficiently thiol alkylates cysteine residues of GSH, leading to its cellular depletion (4) and aggregates with intracellular proteins leading to endoplasmic reticulum (ER) stress (5). The latter causes accumulation of proteins in the ER lumen, thus triggering the UPR activation via the phosphorylation and dimerization of PERK and IRE1α (6). This leads to the activation of eIF2α, JNK1, and P38MAPK, through their phosphorylation, and results in apoptosis via the cleavage of caspases, such as caspase 9, and consequently PARP (7).

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
