# Peer review of "VAS3947 Induces UPR-Mediated Apoptosis through Cysteine Thiol Alkylation in AML Cell Lines"

_ijms, 2020, doi:10.3390/ijms21155470_

Round 1

Reviewer 1 Report

The manuscript submitted by El Dor et al., describes the effect of VAS3947, NOX inhibitor, on the viability and apoptosis in AML cell lines.

The authors show that VAS3947 triggers proliferation arrest and apoptosis. Using HPLC and mass spectrometry, the authors show that VAS3947 thiol alkylates cysteine residues of GSH. VAS3947 also decreased ROS levels . Finally, they demonstrate that VAS3947 triggers the activation of  PERK and IRE1 pathways and the unfolded protein response (UPR).

The manuscript includes interesting data about the effect of NOX inhibitor (VAS3947) on AML. However, in order to improve the manuscript and to be accepted for publication, few points need to be addressed.

Comments:

- Line 20: The authors claim that VAS3947 decreased detectable GSH in MV4-11 and THP-1 cell lines. However, as shown in figure 5D, VAS3947 decreased the GSH level in MV4-11 but not in THP-1.

- Line 23: comma after potential to be removed.

- Figure 3 and 4 : Chromatograms with better image resolution should be included.

-Figure 3C: HPLC with GSH alone should be included.

- Line 141 : Please, include if possible the non shown data in the supplementary materials. Often it is helpful for the reader.

- Line 159-160 : The authors claim a negative correlation between apoptosis and GSH levels. However some cells line have a comparable level of GSH and they show different sensitivity to apoptosis induction by VAS3947.

- Line 178 : « : difference »  to be removed.

- The second paragraph on page 8 of  the manuscript needs to be revised for English editing.

-To confirm the role of GSH depletion on apoptosis, the effect of VAS3947 in combination with N-acetyl cysteine (NAC, a GSH precursor) could be tested.

- Line 203 and line 204 : membrane potential unit is missing.

- Line 207 : the authors refer to figure 6D and 6E to show that NEM deplete glutathione and induce apoptosis. However, Figure 6D and Fig 6E  shows the effect of VAS3947 on mitoROS and total ROS.  The authors should refer instead to Fig 5F and 5G.

- Paragraph 2.4 (page 8) : The authors claim that apoptosis is not triggered by an oxidative stress but rather by aggregation on proteins. Unfortunately, the data provided does not support this statement. The data rather show that VAS3947 reduced the ROS levels.

- Figure 6: Please specify the cell line(s) used in this experiment.

- Did the authors check for the effect of VAS3947 on mitochondrial dysfunction? Example ATP intracellular levels?

- To check if VAS3947 induced apoptosis is due to the disruption of internal ROS balance, the authors could check if the VAS3947 induced apoptosis is rescued by addition of ROS inducers. For instance, with a low concentration of H2O2.

- Quantification of blots relative to loading control, would assist in substantiating claims made especially if sufficient biological replicates were performed.

- Often it is helpful for the reader to see the entire blot to determine specificity of antibodies etc, and protein sizes/variants. This is also becoming a standard supplementary figure requirement in many journals.

- Line 430-442 : The references for the antibodies should be mentioned in the materials and methods.

- Line 235 (page 9) : Replace ‘demonstrates that’  by ‘suggests that’. It would be also interesting to check the effect of ER stress inhibitors on VAS3947 induced apoptosis. These results will potentially support their conclusions.

- Figure 7: Please specify the cell line used in this experiment.

- Line 327: In Figure 3 and 4 GSH and BSA were used at a 10, 20 and 40 µM concentration.

Reviewer 2 Report

In this work, El-Dor and colleagues used AML cell lines as models to investigate the specificity of VAS3947, a NOX inhibitor. They showed that VAS3947 induces apoptosis in AML cells independently of its anti-NOX activity. Using biochemical methods, VAS3947 was shown to thiol alkylate the cysteine residues of glutathione (GSH). Although VAS3947 decreased GSH, suggesting possible oxidative stress induction, however, a decrease of both cytoplasmic and mitochondrial ROS levels was observed. The authors then examined the effect of VAS on ER stress. VAS exposure induced an acute unfolded protein response (UPR) immediately thereafter, through the activation of IRE1a and PERK pathways. Altogether, the authors demonstrated that VAS3947 induces apoptosis independently of its anti-NOX activity, via UPR activation, mainly due to aggregation and misfolding of protein structure.

The study is well performed and interesting. Its originality consists in establishing a new model, and mechanism of action of VAS3947, that has not been shown previously. The paper is well-written, flows smoothly, and well-supported by figures. However, a few minor corrections are suggested that would strengthen the global quality of the paper:

Major comments

  • The NOX inhibitor, VAS3947, shows off-target effects by thiol alkylating cellular proteins, thus targeting AML cell lines by an ER-mediated apoptosis. You have mentioned in the manuscript that VAS3947 is a derivative of VAS2870. However, you do not compare the findings you obtained compared to previous literature about VAS2870. It is best to state a few points regarding how VAS2870 affects ROS and thiol alkylation, which could further support your study.
  • Further supporting your theory of ER-induced apoptosis, it is important that the discussion mentions whether there are any articles that have targeted the ER and detected ROS decrease, similar to your findings or whether this is completely novel mechanism.
  • You have shown that VAS3947 could induce apoptosis by ER stress. It would be interesting to know if the authors have investigated the effect of VAS3947 on autophagy. This could be done by studying the expression of ATG and BECLIN markers.

Minor Comments:

  1. Correct line 26, VAS should be VAS3947, as it is mentioned in the whole manuscript. This should be consistent throughout.
  2. In Figure 2, line 104, please correct VAS to VAS3947, to be consistent with the other figures.
  3. Regarding Figure 5C, line 160, please enlarge the delta Ct annotation. Moreover, it is best to make the genes in italic font.
  4. Lines 200-201, please correct the symbol of mitochondrial membrane potential.
  5. Figure 6, line 207, modify the time to minutes, similar to Figure 7, to maintain consistency.
  6. Lines 226-227, in the results part, it is better to add time points for the different mentioned protein expressions. This makes easier the interpretation of the results and their comparisons.
  7. Line 393, please correct the wavelength symbol.
  8. Figure 8 and line 249, the Figure should be reviewed for some “Style” and “alignment” issues. Words should not touch the boundaries of neighboring words or figures. For example, “Others” should be moved slightly to the right, and so for “mitochondria, cytoplasm, JNK1, and P38”. Moreover, you should improve the alignment of texts to make the figure more organized and better presented.

Round 2

Reviewer 1 Report

The manuscript has been improved.